# Vascular Ageing and Aerobic Exercise

**DOI:** 10.3390/ijerph182010666

**Published:** 2021-10-12

**Authors:** Michaela Kozakova, Carlo Palombo

**Affiliations:** 1Department of Clinical and Experimental Medicine, University of Pisa, 56124 Pisa, Italy; michaela.kozakova@esaote.com; 2Department of Surgical, Medical, Molecular Pathology and Critical Care Medicine, University of Pisa, 56124 Pisa, Italy

**Keywords:** ageing, endothelial dysfunction, arterial stiffness, aerobic exercise

## Abstract

Impairment of vascular function, in particular endothelial dysfunction and large elastic artery stiffening, represents a major link between ageing and cardiovascular risk. Clinical and experimental studies identified numerous mechanisms responsible for age-related decline of endothelial function and arterial compliance. Since most of these mechanisms are related to oxidative stress or low-grade inflammation, strategies that suppress oxidative stress and inflammation could be effective for preventing age-related changes in arterial function. Indeed, aerobic physical activity, which has been shown to improve intracellular redox balance and mitochondrial health and reduce levels of systemic inflammatory markers, also improves endothelial function and arterial distensibility and reduces risk of cardiovascular diseases. The present paper provides a brief overview of processes underlying age-related changes in arterial function, as well as the mechanisms through which aerobic exercise might prevent or interrupt these processes, and thus attenuate vascular ageing.

## 1. Introduction

Chronological age is the major risk factor for cardiovascular (CV) morbidity and mortality. According to the American Heart Association, the prevalence of cardiovascular disease (CVD), including hypertension, coronary heart disease, heart failure and stroke, increases from about 40% in men and women 40–59 years of age, to 70–75% in persons 60–79 years of age, and to 79–86% among those aged 80 years or older [1]. Numerous clinical and experimental studies demonstrated a progressive decline of arterial function with age, even in the absence of conventional CV risk factors and clinical CVD [2,3,4,5,6,7], which gave rise to the theory that arterial dysfunction is a primary effect of advancing age and may represent a link between ageing and CVD [5]. Ageing is also associated with increase in oxidative stress and with low-grade systemic inflammation, both of which play an important role in endothelial dysfunction and arterial stiffening, i.e., in two main age-related arterial phenotypes [8,9]. Consequently, strategies that suppress oxidative stress and inflammation could be effective for preventing age-related changes in arterial function. Physical activity, in general, and regular aerobic exercise, in particular, were shown to diminish oxidative stress and vascular inflammation, improve endothelial function and arterial distensibility and reduce CVD risk [10,11,12,13,14,15]. The present paper provides a brief overview of processes underlying age-related changes in arteries as well as the mechanisms through which aerobic exercise might prevent or interrupt these processes and thus attenuate vascular ageing.

## 2. Endothelial Dysfunction and Ageing

The role of vascular endothelium is to maintain the blood in a fluid state, regulate the vascular permeability, participate in the immune response, adjust the vascular tone to changes in blood flow and generate a new vascular network or repair an existent one. To fulfill this complex role, a healthy endothelium must constantly maintain a balance between prothrombotic and antithrombotic signals, proinflammatory and anti-inflammatory molecules, oxidants and antioxidants, vasodilators and vasoconstrictors. Endothelial dysfunction occurs when prevail prothrombotic, proinflammatory, oxidant or vasoconstrictor features.

Endothelial homeostasis is maintained mainly through the release of vasoprotective molecule of nitric oxide (NO) that is synthetized by endothelial nitric oxide synthase (eNOS) from L-arginine, in the presence of molecular oxygen, reduced nicotinamide-adenine-dinucleotide phosphate (NADPH) and cofactor tetrahydrobiopterin [16,17,18]. Any condition associated with impaired NO synthesis or excessive NO degradation may cause endothelial dysfunction.

A primary mechanism responsible for age-related endothelial dysfunction is oxidative stress characterized by an excessive production of reactive oxygen species (ROS) and/or a reduced capability of endogenous antioxidant defense system (Figure 1). Excessive ROS within endothelial cells react directly with NO leading to its deactivation and to the formation of another ROS, peroxynitrite. ROS also oxidize tetrahydrobiopterin, thus inhibiting NO synthesis and contributing to eNOS uncoupling with subsequent production of superoxide instead of NO. In an experimental study on mesenteric arteries of ageing mice, a treatment with exogenous tetrahydrobiopterin increased endothelium-dependent vasodilation and reduced eNOS-derived superoxide formation, confirming an important role of tetrahydrobiopterin and e-NOS uncoupling in age-related endothelial dysfunction [18,19].

With increasing age, the production of ROS by mitochondria increases, as well as the activity of NADPH oxidases, whereas the expression and activity of endogenous antioxidant enzymes like superoxide dismutase, catalase and glutathione decreases. In mitochondria, ROS are generated as by-products of aerobic respiration, and therefore, mitochondria are the major producer of ROS in cells. Electrons leaking from the electron transport chain to oxygen produce short-lived free radicals such as O_2_^·−^ that can be converted to more stable and membrane-permeable H_2_O_2_. Mitochondrial ROS interact with lipids, proteins and nucleic acids and cause their oxidative damage. With ageing these oxidative damages accumulate and result in impairment of mitochondrial morphology and function. Oxidative damage is more critical in specific molecular targets, such as mitochondrial DNA (mtDNA) that encodes essential components of oxidative phosphorylation and protein synthesis. Oxidative damage of mtDNA deteriorates the function of the respiratory chain and triggers further accumulation of ROS [20,21]. An impact of age on ROS production as well as an impact of ROS on mitochondrial function were confirmed in experimental studies. Endothelial cells of older mice had significantly higher mitochondrial O_2_^·−^ production and increased mitochondria-derived cellular H_2_O_2_ levels as compared to endothelial cells of younger mice. A treatment of human umbilical vein endothelial cells with reactive species (H_2_O_2_ and ONOO^−^) was associated with an increase in mtDNA damage and decrease in mitochondrial RNA transcripts, protein synthesis, ATP levels and redox potential [22,23].

Another important source of ROS is NADPH oxidase, which converts molecular O_2_ into O_2_^·−^ that can be further converted to H_2_O_2_ or OH^·−^ or peroxinitrite. In ageing endothelial cells, the expression of isoform 4 of NADPH oxidase family (Nox4) increases, which contributes to ROS accumulation within the cell and to an increase in mitochondrial ROS levels if the functional Nox4 is also present in mitochondria [24,25].

Endothelium has an inherent ability to repair its damage and restore its function by mobilizing endothelial progenitor cells (EPCs) from bone marrow and other tissues. Healthy EPCs express high levels of antioxidant enzymes, and as a result, are more resistant to oxidative stress [26]. With ageing declines not only the number of EPCs but also their antioxidant capacity, and thus their regenerative potential [27,28].

A hallmark of ageing process is a low-grade inflammation (“inflamm-aging”). In the aged vascular wall a cross-talk takes place between increased oxidative stress and activation of inflammatory processes. ROS act as signaling molecules and activate nuclear factor kappa-light-chain-enhancer of activated B cells (NF-κB) in endothelial cells and SMCs. Activated NF-κB induces the transcription of a large number of genes implicated in vascular inflammation, like adhesion molecules, cytokines, and chemokines, and it also stimulates ROS production via activation of NADPH oxidase [29,30,31]. The expression of NF-κB, interleukin (IL)-6 and monocyte chemoattractant protein-1 (MCP-1) was found to be higher in endothelial cells of older healthy adults as compared to younger ones [32], and a suppression of NF-κB signaling in endothelial cells of older obese adults increased endothelium-dependent vasodilation by 74% [33].

## 3. Endothelial Dysfunction and Aerobic Exercise

Chronic aerobic exercise is supposed to prevent and/or reverse age-related endothelial dysfunction by increasing NO bioavailability, improving intracellular redox balance and mitochondrial health, increasing number and recovering capability of circulating EPCs and lowering systemic inflammation (Figure 1).

An important physiologic stimulus to endothelial NO production is an increase in intravascular blood flow as the endothelium tends to normalize shear stress through NO-mediated vasodilation [34]. Exposure of endothelial cells to unidirectional laminar shear acutely stimulates production of NO by eNOS and it also stimulates production of tetrahydrobiopterin in order to assure the function of eNOS enzyme [35]. This shear-stress-mediated effects may explain the observation that a single session of exercise elicited an acute increase in NO formation, both in athletes and healthy non-athletic subjects [36]. In experimental studies, a 2-to 4-week exercise training in rats increased NO synthesis and improved flow-mediated dilatation in skeletal muscle arteries [37], increased eNOS protein levels in aortic tissue and enhanced acetylcholine-induced aortic vasodilatation [37,38]. As expected, shear-stress-mediated effect on NO production can be detected only in arterial districts subjected to exercise hyperemia. A 4-week exercise in rats improved flow-mediated dilatation in skeletal muscle arteries but not in mesenteric arteries [39].

Experimental studies in healthy aged animals demonstrated that aerobic exercise improves mitochondrial biogenesis and upregulates mitochondrial antioxidant defense system [40]. In large arteries of aged rats and mice, aerobic exercise training suppressed mitochondrial ROS formation and NADPH oxidase protein expression, increased expression of antioxidant enzymes and mtDNA content, enhanced ATP formation and reduced mitochondrial swelling [41,42,43]. Observed changes in mitochondrial health were accompanied by improvement in endothelium-mediated arterial relaxation.

Aerobic exercise also stimulates the mobilization of EPCs. In healthy individuals, a single bout of exercise induced a release of EPCs from bone marrow into the circulation for up to 72 h post-exercise [44]. The mechanisms by which exercise mobilizes EPCs is complex and includes activation of vascular endothelial growth factor (VEGF) that in turn activates matrix-metalloproteinase (MMP)-9, which stimulates stem cells to migrate from a quiescent bone marrow niche to the vascular zone [45]. Indeed, a 4-week running program in mice resulted in an upregulation of VEGF serum levels, increase in the number of circulating EPCs and augmented EPCs production within the bone marrow [46].

The response of vascular inflammatory markers to aerobic exercise depends on exercise duration and intensity. Although a single bout of exercise increased circulating levels of IL-6, -8, -10, tumor necrosis factor-α (TNF-α) and C-reactive protein (CRP) [47,48], regular aerobic exercise was associated with decrease in serum concentration of inflammatory markers. A 2-week voluntary wheel running of old mice reduced activation of the proinflammatory transcription factor NF-κB, diminished aortic expression of the proinflammatory cytokines IL-1 and IL-6 and ameliorated NO-mediated endothelium-dependent dilation [49], and an 8-week running protocol in obese rats decreased plasma TNF-α, increased e-NOS protein expression and improved endothelium-dependent relaxation [50].

Studies in humans provide additional evidence that regular aerobic exercise prevents the age-associated loss in endothelium-dependent vasodilation and improves endothelium function in sedentary individuals. In middle-aged and older healthy sedentary men the forearm blood flow response to acetylcholine was 25% lower as compared to young sedentary men, while there were no differences in blood flow response between younger and older endurance-trained men. Moreover, a 3-month, home-based aerobic exercise in older sedentary men increased acetylcholine-mediated vasodilation by 30% [51]. Older exercising men had lower endothelial expression of NADPH oxidase and NF-κB, higher expression of the antioxidant enzymes and greater brachial flow-mediated dilation (FMD) as compared to older sedentary men [52]. Aerobic exercise in obese women decreased mitochondrial H_2_O_2_ production, increased catalase antioxidant activity and decreased DNA oxidative damage [53]. A 3-month aerobic exercise in older men improved EPCs number and function nearly 2-fold [54], and 12-week exercise markedly enhanced reendothelialization capacity of EPCs from elderly men [55].

## 4. Arterial Stiffening and Ageing

Arterial compliance describes the ability of large elastic arteries to expand during systole and recoil during diastole, which guarantees an optimal transport of blood from the left ventricle to the periphery during entire cardiac cycle. Loss of arterial compliance reduces diastolic pressure, increases flow pulsatility along the arterial tree and augments the propagation velocity of incident waveforms, thus anticipating the return of reflected waveforms to systole with consequent increment in systolic blood pressure and pulse pressure. Described changes compromise coronary perfusion, augment afterload and cardiac workload and trigger endothelial dysfunction [56,57,58].

Extracellular matrix (ECM), above all its structural proteins elastin and collagen, provides large arteries with elasticity and strength that are necessary for their proper functioning. Elastin, the main component of elastic fibers, allows the large arteries to expand and relax with every cardiac cycle, while collagen represents the load-bearing and reinforcing element of arterial wall [59]. Yet elastic fibers have an extremely low turnover rate and this longevity results in accumulation of damages caused by mechanical fatigue, which leads to elastin fragmentation and loss of arterial compliance. In ageing vessels, the absolute amount of elastin decreases and that of collagen increases, shifting the collagen-elastin balance that governs an appropriate arterial function [60].

Stability of ECM is controlled by MMPs, a group of endopeptidases capable to degrade collagen, elastin and other extracellular molecules (Figure 2). With ageing the activity of MMP-2/-7/-9/-14 [61,62] increases, i.e., the activity of MMPs able to break down elastin, and in case of MMP-2/-9 also collagen [63]. Collagen and other ECM proteins of ageing vessels can be modified by advanced glycation end products (AGEs). Since formation of AGEs is a slow process that affects proteins with long half-lives that contain exposed lysine residues, fibrillar collagen is highly susceptible to glycation and cross-linking. Intermolecular AGEs-mediated cross-linking of collagen increases arterial stiffness and reduces its mechanical robustness [64].

Inflamm-aging also participates on arterial stiffening. The chronic proinflammatory profile within ageing arteries is characterized by alterations in major signaling cascades that include the renin/angiotensin II and the endothelin-1 (ET-1) /endothelin-1 receptor systems, and is mainly generated by endothelial cells and SMCs [65]. Angiotensin II or ET-1 activate MCP-1, transforming growth factor-β1 (TGF-β1), IL-2/-6/-17 and MMP-2 expression in vascular SMCs, induce SMC proliferation, migration and ECM synthesis [65,66,67]. Transition of vascular SMCs from the contractile to the synthetic phenotype results in arterial wall thickening which has been shown to be associated with lower distensibility [68,69].

Arterial compliance had two components, a passive component that reflects artery wall composition, and an active component related to vascular tone. The major locally released vasoactive substances modulating vascular tone are NO and ET-1 [70,71]. NO initiates and maintains vasodilation through a cascade of biological events that culminate in the relaxation of SMCs. ET-1 exerts its actions on vascular SMCs by binding to two specific receptors, the endothelin-A (ET_A_) and endothelin-B (ET_B_) receptors. ET_A_ receptor is the primary receptor driving the potent vasoconstrictor effects of ET-1 in healthy vasculature. The expression of ET_B_ receptors in vascular SMCs is low, but ET_B_ receptors are also located on endothelial cells, where they promote vasodilation via the release of NO and prostacyclin [72,73,74].

Age-related changes in ECM, MMPs activity, vascular inflammation, and arterial stiffness were demonstrated in animal models as well as in ageing humans. A two-fold increase in the collagen content together with SMC hypertrophy, luminal enlargement and lower compliance were observed in the aortas of 30-month old rats when compared to 6-month old rats [75]. In human aortic tissue, the concentration of collagen increased by 72% over the age 14–90 years [76], while the amount of interlaminar elastic fibers with age decreased [77]. In ageing mice, the increase in carotid stiffness was associated with increases in collagen I and III in the adventitia and with reduction of elastin in the media, in which there was also an increase in MMP-2 activity [78]. MMP-2 activity increased progressively with age in the aortic intima and media of rats [79] and in human aortic tissue [80], and both MMP-2 activity and angiotensin II expression were increased, 3-fold and 5-fold respectively, in aortas of old male monkeys when compared to aortas of young monkeys [81]. A chronic administration of MMP inhibitor to 16-month–old rats blunted the age-associated increases in aortic MMP-2 activity, diminished the elastin fiber degeneration and collagen deposition, and reduced the activation of MCP-1 [82]. In normotensive rats, the treatment with AGEs inhibitor prevented the age-related increase in aortic impedance in the absence of changes in collagen and elastin content, an observation suggesting that the effect of treatment was related to a decrease in the AGEs-mediated cross-linking of ECM [83]. In a large prospective human study, circulating soluble receptors for AGEs as well as skin-deposed AGEs were independently associated with a yearly increase in aortic pulse wave velocity (PWV) [84]. Increased levels of angiotensin converting enzyme, angiotensin II, angiotensin receptor type 1 MMPs and MCP-1 were found in aged human aortic wall when compared to young aortas [85], and a 12-month administration of angiotensin II receptor blocker to elderly hypertensive patients reduced aortic PWV, independently of blood pressure changes [86]. Finally, in aging healthy women plasma levels of ET-1 significantly increased [87] while the expression of endothelial ET_B_ receptor decreased [88], and healthy older men had increased ET-1 expression in vascular endothelial cells [89].

## 5. Arterial Stiffening and Aerobic Exercise

Experimental and human studies identified the mechanisms by which aerobic exercise prevents and/or reverses age-related arterial stiffening. These mechanisms include reduction in MMPs activity and AGEs accumulation, inhibition of SMCs proliferation and migration and decrease in oxidative stress and inflammation (Figure 2).

A 10-week voluntary wheel running in old mice reversed age-related increase in aortic PWV, decreased aortic content of collagen-I and AGEs to the levels observed in young mice, and reduced aortic superoxide and nitrotyrosine levels by 50% [90]. A 2-week modest voluntary wheel running in old mice improved carotid stiffness and reduced content of collagen I, collagen III and TGF-β1 in carotid tissue [78]. A 5-month exercise in mice with Marfan syndrome reduced MMP-2 and MMP-9 expression in aortic tissue, decreased elastic fiber fragmentation in aortic wall, diminished aortic diameter and improved aortic wall elasticity [91]. An 8-week period of swimming in male mice reduced aortic collagen and AGEs content and improved aortic PWV and subendothelial matrix stiffness. However, after 8 weeks of rest, the values returned to the baseline, suggesting that the beneficial effect of exercise is short-lived [92]. A 17-week training in aortic-banded mini-swine prevented coronary artery stiffening, preserved arterial elastin content, avoided arterial AGE accumulation and cytokine release from perivascular adipose tissue [93]. Voluntary wheel running in older mice reduced aortic expression of NF-κB and proinflammatory cytokines [49] as well as aortic infiltration by T cells and macrophages [94]. Moreover, exercise-induced increase in intravascular blood flow and shear stress inhibited SMCs proliferation and neointimal thickening [95].

The positive impact of aerobic exercise on arterial stiffness was confirmed also in humans. In healthy volunteers, a higher aerobic capacity (higher maximal oxygen consumption) was associated with reduced arterial stiffness [96], and in a large population-based sample of older adults, habitual physical activity from mid- to late-life was associated with lower aortic PWV and lower central pulse pressure [97]. In subjects with metabolic syndrome, exercise reduced aortic PWV as well as plasma MMP-1 and MMP-7 [98]. In contrast, no differences in plasma activity of MMP-2, MMP-7, and MMP-9 were observed between sedentary subjects and subjects engaged in lifelong exercise, despite differences in central arterial stiffness [99]. A 12-week moderate-intensity walking exercise in diabetic patients reduced plasma levels of angiotensin II and AGEs, and changes in AGEs correlated with changes in aortic PWV [100]. In healthy older adults, habitual exercise decreased expression of proinflammatory/-oxidative genes in peripheral blood mononuclear cells [101], and in a large prospective study, subjects adhering to standard physical activity recommendations for cardiovascular health displayed lower CRP and IL-6 levels [102]. A 3-month regular aerobic exercise decreased plasma ET-1 levels in older women [87] and a 12-week aerobic training of middle-aged and older adults increased carotid compliance, reduced plasma ET-1 concentration and eliminated ET-1-mediated vascular tone [103].

## 6. Aerobic Versus Resistance Exercise

As demonstrated by numerous studies discussed above, aerobic exercise prevents and reverses age-related endothelial dysfunction and arterial stiffening, yet it exerts limited impact on muscle mass and strength and on osteoporosis. In fact, an age-related loss in muscle strength in older adults who perform chronic endurance training alone is similar to that of sedentary adults [104]. Therefore, aerobic exercise should be combined with resistance exercise that increases muscle strength, cross-sectional area of muscle fibers and bone density [105,106], though its effect on arterial function is controversial.

A single session of moderate intensity resistance exercise in middle-aged sedentary men increased plasma levels of nitrites and nitrates and improved FMD, while a single session of high intensity resistance exercise increased plasma levels of ET-1 and did not improve FMD [107]. However, an 8-week resistance exercise training reduced the plasma ET-1 levels [108]. In overweight women with type 2 diabetes mellitus, aerobic but not resistance exercise improved FMD, though the decrease in body mass index was similar with the two exercise modalities [109]. In contrast, in subjects with hypertension and in patients with recent acute myocardial infarction, aerobic and resistance exercise improved endothelial function to a comparable extent [110,111]. Some studies also tested the effect of combined aerobic and resistance exercise on endothelial function. In young healthy subjects, impaired endothelial function after the resistance exercise was restored with successive 10-min cycling [112], while 45-min cycling prior to resistance exercise did not prevent the acute impairment in endothelial function after resistance exercise [113]. The detrimental effects of resistance exercise on endothelial function in young healthy subjects was also prevented when resistance exercise was performed at high intensity but with low repetitions [114].

The results of studies assessing the impact of resistance exercise on arterial stiffness are also inconsistent. One bout of resistance exercise in healthy volunteers acutely decreased carotid compliance and increased carotid-femoral PWV that, however, returned to baseline values in less than 1 h after exercise [115,116]. An 11-week high-intensity resistance training in young healthy women increased carotid-femoral PWV and augmentation index [117]. A 4-week resistance training in prehypertensive subjects increased aortic PWV, while aerobic training of the same duration aortic PWV decreased [118]. Yet, in young trained men, resistance training, either with heavier load/lower repetition or lighter load/higher repetition, reduced central arterial stiffness [119], and in healthy middle-aged men, five stretching sessions per week for 4 weeks reduced brachial-ankle PWV [120].

Altogether, available data indicate that regular aerobic exercise improves aerobic capacity and has a favorable effect on vascular ageing and CV risk. Resistance exercise improves muscle strength and maintain bone mineral density, while its effect on arterial stiffness and endothelial function might depend on the load and repetition frequency. Therefore, a well-balanced combination of aerobic and resistance exercise could improve general well-being in elderly.

## 7. Conclusions

Deterioration of vascular function represents a major link between ageing and CV risk, given that the two most important age-related arterial phenotypes, endothelial dysfunction and large elastic artery stiffening, are strong independent predictors of CVD. Although the endothelial dysfunction and large artery stiffening are two distinct pathophysiologic entities, they have common underlying mechanisms, like oxidative stress and low-grade inflammation, and may exacerbate each other, thus enhancing the vascular dysfunction with advancing age. Numerous molecular mechanisms activating oxidative stress or chronic vascular inflammation can be controlled by aerobic exercise (Figure 1 and 2), and consequently, regular aerobic exercise is considered an optimal strategy for preventing or reversing age-related degenerative vascular changes and reducing risk of CVD.

A beneficial effect of exercise on CV system can be partially explained also by the exercise-induced changes in glucose metabolism and lipid profile [121]. Exercise-stimulated glucose uptake by muscle occurs independently of insulin signal transduction, making exercise an excellent nonpharmacological method for reducing hyperglycaemia in insulin-resistant conditions, like obesity and type 2 diabetes mellitus. In addition, exercise promotes a short-time increase in insulin sensitivity after the cessation of physical exertion, which, after 60 min of moderate ergometer cycling, can last up to 48 h in healthy volunteers and up to 15 h in patients with type 2 diabetes mellitus [122,123]. Physical activity level has been also shown to be associated with plasma triglycerides and HDL-cholesterol levels, inversely and directly, respectively [124].

According to guidelines of the European Society of Cardiology [125] as well as according to Physical Activity Guidelines for Americans [126], healthy adults of all ages should perform at least 150 min to 300 min a week of moderate-intensity aerobic physical activity or 75 min to 150 min a week of vigorous-intensity aerobic physical activity or combination thereof. Adults should also do muscle-strengthening activities of moderate or greater intensity that involve all major muscle groups on 2 or more days a week.

Food supplementation with substances having antioxidant and antiinflammatory properties, like vitamin C, vitamin E and thiols, might further improve endothelial function and arterial stiffness in elderly [127,128,129,130,131,132].

## Figures and Tables

**Figure 1 ijerph-18-10666-f001:**
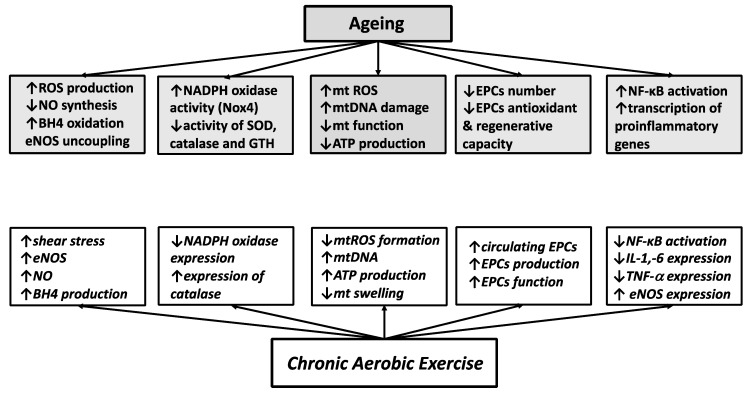
Mechanisms through which ageing deteriorates endothelial function (grey rectangles) and mechanisms through which chronic aerobic exercise improves endothelial dysfunction (white rectangles). ROS: reactive oxygen species; NO: nitric oxide; BH4—tetrahydrobiopterin; eNOS: endothelial nitric oxide synthase; NADPH: nicotinamide-adenine-dinucleotide phosphate; SOD: superoxide dismutase; GTH: glutathione; mt: mitochondrial; EPCs: endothelial progenitor cells; NF-κB: nuclear factor kappa-light-chain-enhancer of activated B cells; IL: interleukin; TNF-α: tumor necrosis factor-α.

**Figure 2 ijerph-18-10666-f002:**
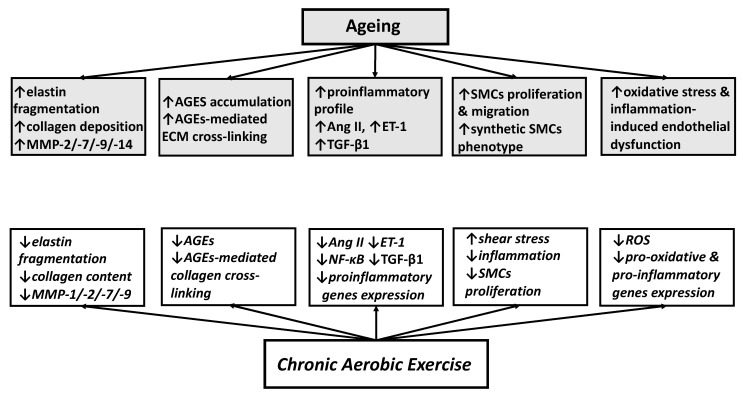
Mechanisms through which ageing deteriorates arterial compliance (grey rectangles) and mechanisms through which chronic aerobic exercise improves arterial compliance (white rectangles). MMP: matrix-metalloproteinase; AGEs: advanced glycation end products; Ang II: angiotensin II; ET-1: endothelin-1; NF-κB: nuclear factor kappa-light-chain-enhancer of activated B cells; TGF-β1: transforming growth factor-β1; SMCs: smooth muscle cells; ROS: reactive oxygen species.

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
