# Peer review of "Vascular Ageing and Aerobic Exercise"

_ijerph, 2021, doi:10.3390/ijerph182010666_

Round 1

Reviewer 1 Report

This review focuses on the changes associated with vascular ageing in relation to endothelial dysfunction and arterial stiffening (in large arteries). By outlining the associated processes leading an increased prevalence of cardiovascular disease, the review then subsequently overlays the benefits of aerobic exercise in preventing these changes from occurring. In addition, a brief comparison between the benefits of aerobic versus resistance exercise in these processes is presented.

Specific comments:

This reviewer found the figures somewhat confusing at first. The arrows from chronic aerobic exercise appear to feed into the associated pathologies occurring with ageing. While this clearly is not the intention of the article, this is the impression from figures 1 and 2.

In the “Endothelial dysfunction and aerobic exercise” section, paragraph 2 indicates NO is increased with a single bout of exercise, but in paragraph 5 a single bout of exercise leads to elevated inflammatory markers. These appear to be in opposition and the bigger question would be at what point does “regular” exercise become beneficial?

Has any assessment of endothelin receptor subtype expression been completed during the ageing process and does this contribute to the changes associated with ageing?

The comparison of aerobic versus resistance exercise may be beyond the scope of this review. Since there are an infinite number of protocols for resistance exercise and given the variability in approaches is quite difficult to ascertain the impact of resistance training overall on either endothelial dysfunction or arterial stiffening.

The recommendations for exercise in the conclusions section only take into account those from the European Society of Cardiology, but there are other recommendations that include resistance training as an essential component for overall health (i.e. the US Guidelines).

Author Response

1) The figures have been simplified and made more clear to understand.

2) We reported data from different studies that not always go in the same direction. In this case it seems that a single bout of exercise elicits increased NO production through shear, but it also increases circulating levels of inflammatory markers. On the other side, a regular exercise seems to decrease serum concentration of inflammatory markers. We rephrased the last sentence to highlight the difference between a single bout and regular exercise (see lines 151-154).

3) Thank you for the comment. We have added a small paragraph on the role of endothelin in arterial stiffening. We also reported some studies regarding endothelin and ageing or endothelin and aerobic exercise.  (see lines 216-225, 250-252 and 288-291; reference 70-75, 87-89, 103).

4) The review is focused on impact of aerobic exercise on age-dependent endothelial dysfunction and arterial stiffening. However, we think that resistance exercise should be briefly mentioned since it could provide additional benefits in elderly, especially for muscle strength and bone density. Therefore the combination of both modalities should be considered. This concept is now better expressed. (see lines 294-300; reference 104-106).

4) In the conclusions we now report also Physical Activity Guidelines for Americans and their recommendation on resistance training. (see lines 355-361; reference 126).

All changes are highlighted in yellow.

Reviewer 2 Report

In the manuscript “Vascular Ageing and Aerobic Exercise”, the authors have summarized evidence about processes underlying age-related changes in arterial function, as well as the mechanisms through which aerobic exercise might prevent or interrupt these processes, and thus attenuate vascular ageing. The topic of the manuscript is certainly interesting and captures one of the questions of the moment on cardiovascular prevention.), but not when levels are less than the median.

The manuscript is well-written; however, I have some minor comments:

  • Please reformulate the figures to make them clearer. The manuscript is well written, the figures should be just as illustrative.
  • Please, briefly discuss the biochemical and hematological markers that vary significantly during and after sports training to identify risk factors, as reported in recent review (ref. Clin Chem Lab Med. 2019 Sep 25;57(10):1450-1473.doi: 10.1515/cclm-2018-1107.)
  • Do the authors believe it is beneficial to use antioxidant supplements in patients who exercise? Please discuss this point briefly (ref. Int J Environ Res Public Health. 2020 Dec 16;17(24):9424.doi: 10.3390/ijerph17249424).
  • Please, you should explain each of your abbreviations in the figure caption.

Author Response

1) The figures have been simplified and made more clear to understand. All abbreviations in the figure caption are now explained.

2) Thank you for the suggestion. We added a short paragraph regarding the impact of exercise on glucose uptake and lipid profile, since it is directly related to CV prevention in elderly. On the other hand, we do not discuss hematologic markers, as the question is more complex, concerns above all the elite athletes, and thus, it is not directly associated with vascular ageing (see lines 345-354; ref. 121-124).

3) The possible benefit of substances with anti-oxidant and anti-inflammatory properties is briefly mentioned in conclusions together with pertinent references (lines 362-364, ref. 127-132).

All changes are highlighted in yellow.